# Beyond Antivirals: Alternative Therapies for Long COVID

**DOI:** 10.3390/v16111795

**Published:** 2024-11-19

**Authors:** Achilleas Livieratos, Charalambos Gogos, Karolina Akinosoglou

**Affiliations:** 1Independent Researcher, 15238 Athens, Greece; 2Department of Medicine, University of Patras, 26504 Rio, Greece; cgogos@med.upatras.gr (C.G.); akin@upatras.gr (K.A.); 3Department of Internal Medicine and Infectious Diseases, University General Hospital of Patras, 26504 Rio, Greece

**Keywords:** long COVID, post-COVID condition, non-antiviral treatments, SARS-CoV-2

## Abstract

Long COVID or Post-Acute Sequelae of SARS-CoV-2 infection (PASC) is a condition characterized by numerous lingering symptoms that persist for weeks to months following the viral illness. While treatment for PASC is still evolving, several therapeutic approaches beyond traditional antiviral therapies are being investigated, such as immune-modulating agents, anti-inflammatory drugs, and various supportive interventions focusing at alleviating symptoms and enhancing recovery. We aimed to summarize the breadth of available evidence, identify knowledge gaps, and highlight promising non-antiviral therapies for Long COVID/PASC. We followed the framework of a scoping methodology by mapping existing evidence from a range of studies, including randomized clinical trials, observational research, and case series. Treatments evaluated include metformin, low-dose naltrexone (LDN), dexamethasone, statins, omega-3 fatty acids, L-arginine, and emerging therapies like intravenous immunoglobulin (IVIg) and therapeutic apheresis. Early findings suggest that metformin has the strongest clinical evidence, particularly from large phase 3 trials, while LDN and dexamethasone show potential based on observational studies. However, many treatments lack robust, large-scale trials. This review emphasizes the need for further research to confirm the efficacy of these treatments and guide clinical practice for Long COVID management.

## 1. Introduction

Many patients who have managed to recover from COVID-19 continue to face lingering symptoms well past the acute phase of the initial infection [1,2]. Long COVID, defined as a postviral syndrome, persists for a minimum of 3 months following the viral illness [1]. It may present in various forms, including ongoing, recurrent, or progressively worsening symptoms, and can have systemic implications [1]. Symptoms include fatigue, cognitive impairment, shortness of breath, and a wide range of other systemic issues [1]. Long COVID, affecting an estimated 10% to 30% of those infected, can occur in anyone—including individuals with mild or asymptomatic cases—though it is more commonly associated with a severe initial illness [1,2]. The condition may also exacerbate pre-existing health problems and lead to significant functional impairment [1]. These symptoms have not only impeded recovery for millions, but have also burdened healthcare systems globally, driving the urgent need for effective therapeutic interventions [2,3]. Among these symptoms, neuro-Long COVID, characterized by cognitive difficulties and fatigue—commonly referred to as “brain fog”—is a prevalent and persistent aspect of Long COVID-19 syndrome [4,5,6,7]. Approximately up to 30% of patients experience cognitive impairments, with nearly 46% reporting some type of memory disruption [4,5,6,7]. The persistence of these cognitive issues, lasting for months or even years, underscores the necessity for long-term follow-up and specialized interventions to manage the enduring neurological impacts of Long COVID.

Long COVID risk also varies across different populations, with certain groups facing higher vulnerability [4]. Special populations at increased risk include the elderly, individuals with various co-morbidities, and immunocompromised patients [4]. These groups often experience more severe symptoms and longer-lasting effects [4]. Additionally, women and individuals with a history of autoimmune diseases may be more susceptible to developing Long COVID symptoms, including fatigue, cognitive dysfunction, and musculoskeletal pain [4,5,6]. The persistent inflammation and immune dysregulation that accompany Long COVID appear to be independent of the SARS-CoV-2 variant, suggesting that, while the severity of acute illness may vary, the risk of developing Long COVID persists across all variants [4]. Finally, the impact of vaccination on Long COVID risk is significant as there is reduced risk of developing Long COVID compared to unvaccinated individuals [5]. Immunization reduces both the severity of acute infection as well as the probability of experiencing prolonged symptoms [5].

Long COVID appears to be a multifactorial syndrome, with several mechanisms potentially contributing to its complex pathology [1,7]. Hypotheses under investigation include immune dysregulation, mitochondrial dysfunction, endothelial damage, persistent inflammation, autoimmunity, viral persistence, dysautonomia, and microbe dysregulation or dysbiosis [7]. Chronic immune system alterations are a key feature of long COVID, with persistent activation of innate immune cells, exhausted T cells, and elevated levels of interferons (type I and III) lasting months after infection [8,9,10]. This immune dysregulation suggests the body may fail to return to homeostasis post-infection, leading to prolonged symptoms [8,9,10]. Mitochondrial dysfunction, likely driven by dormant viruses such as Epstein–Barr Virus (EBV), results in impaired energy metabolism, chronic fatigue, and exercise intolerance [11,12,13,14,15]. In addition, endothelial dysfunction contributes to microvascular blood clotting and impaired oxygen delivery, which may increase the risk of long-term cardiovascular complications like deep vein thrombosis and pulmonary embolism [16].

A persistent inflammatory response, characterized by elevated pro-inflammatory cytokines such as TNF, mirrors other conditions like myalgic encephalomyelitis/chronic fatigue syndrome (ME/CFS) and suggests that inflammation continues long after the acute phase of infection [7]. Moreover, the presence of elevated autoantibodies targeting receptors such as ACE2 and β2-adrenoceptors indicates that autoimmunity, possibly triggered by molecular mimicry during SARS-CoV-2 infection, could be contributing to the prolonged symptoms [17,18]. Viral persistence is another significant hypothesis, as SARS-CoV-2 proteins or RNA have been found in various tissues, potentially maintaining a reservoir that drives long-term symptoms [19,20].

Dysautonomia, often manifesting as postural orthostatic tachycardia syndrome (POTS), is frequently observed in these patients and may result from immune system dysregulation or direct viral effects on the autonomic nervous system, leading to symptoms such as dizziness, abnormal heart rate, and fatigue [21]. Finally, gut microbiota disruption, or dysbiosis, has been noted in Long COVID, with reduced levels of beneficial bacteria and an increase in harmful species, contributing to both gastrointestinal and systemic symptoms, including fatigue and cognitive impairment [22]. These multiple overlapping mechanisms highlight the complexity of Long COVID and its far-reaching impact on various organ systems.

Findings indicate that antiviral treatment, when administered early (within 5 days of symptom onset), significantly reduces the occurrence of PASC by 27.5% [23]. Moreover, antiviral therapy also lowers the risk of PASC-related hospitalizations and death by 29.7% [23]. This reduction suggests that antiviral drugs may effectively prevent the long-term persistence of the virus in the body, which has been hypothesized to contribute to the ongoing symptoms associated with Long COVID [23,24]. Additionally, the use of antiviral drugs helps to reduce viral load, which could mitigate the prolonged inflammation and immune dysregulation seen in many Long COVID patients [23,24]. However, while these results are promising, non-antiviral therapies that target the previously identified molecular pathways have emerged also as potential treatment strategies [2,3,25,26,27,28,29,30,31,32,33,34,35,36,37,38,39,40,41].

This review aims to synthesize the current evidence on non-antiviral treatments for Long COVID, highlighting potential therapeutic options and identifying areas where further research is urgently needed. Specifically, the goal is to evaluate the efficacy of different treatments in addressing the multifaceted symptoms of Long COVID based on their clinical impact and feasibility of implementation. By examining and analyzing the available data, this study seeks to guide future clinical trials and provide insights for future evidence-based treatment guidelines for Long COVID, offering hope to the millions of patients suffering from this debilitating condition.

## 2. Materials and Methods

Between January 2021 and September 2024, a comprehensive literature review was performed using PubMed, which initially identified 2032 potentially relevant articles. The search involved the following query: COVID-19 treatment OR Management AND Long COVID AND Post-acute sequela of SARS-CoV-2 (PASC) AND Post-COVID-19 condition (PCC) OR Syndrome. 1900 papers were excluded for being out of scope or involving animal research. Only English-language studies were considered, with duplicates removed. A detailed evaluation of the remaining 132 articles led to the removal of 114 studies due to their focus on anti-viral or combination treatment. Ultimately, 18 studies met the inclusion criteria (Figure 1). Two researchers reviewed and hand-searched the literature, resolving any disagreements collaboratively.

Among the 18 articles evaluated, design and patient number variations were significant between different studies. Collectively, these articles aimed to capture the diversity of different non-antiviral treatments investigated for Long COVID patients. Furthermore, advanced, in-progress pharmacotherapy trials (n = 7) for Long COVID patients are listed in a separate table. This includes phase 3–4 trials, with estimated trial completion dates prior to 2026, and the following search strings were applied when searching the Clinical Trials platform: Long COVID, Post-COVID Syndrome, Post-COVID Condition, and Post-acute sequela of SARS-CoV-2.

## 3. Results

In total, 18 studies were identified across seven therapeutic categories regarding available treatments (Table 1). The Antihypertensive/ADHD Treatment category included one study, an open-label clinical experience involving guanfacine. The Antioxidant category included one phase 2 RCT and one observational study involving Coenzyme Q10. The Antidiabetic category comprised two phase 3 RCTs investigating metformin for reducing Long COVID risk. The Immunomodulator category had the largest number of studies, with eight in total, focusing on drugs such as low-dose naltrexone (LDN), dexamethasone, and intravenous immunoglobulins (IVIg), primarily through retrospective observational and case–control studies. The Statin category featured one observational study investigating atorvastatin/rosuvastatin. Lastly, the Other category included four observational studies examining interventions like omega-3, L-arginine + vitamin C, and therapeutic apheresis.

Among the ongoing trials, seven studies were identified across various therapeutic categories targeting Long COVID (Table 2). The Immunomodulator category includes three studies: Anakinra, Ibudilast, Pentoxifylline, and Infliximab. These are primarily phase 2–3 randomized controlled trials (RCTs), except for the Infliximab study, which is a phase 4 randomized open-label trial. The Antidepressant category includes two studies: Fluvoxamine, and Fluvoxamine or Metformin, both of which are Phase 2–3 RCTs. The Other category includes two studies: one on ImmuneRecov, a Phase 3 non-randomized open label trial, and one on Testofen, a phase 3 RCT investigating its effects on energy and fatigue.

## 4. Discussion

Among the available treatments, metformin stands out with the strongest clinical evidence, due to two large phase 3 trials, which demonstrated substantial reductions in the incidence of Long COVID (42% to 63%) [26,27]. The high-quality design and consistent outcomes across large patient populations make metformin a leading candidate for preventing Long COVID. LDN also shows strong evidence, particularly in improving symptoms like fatigue, post-exertional malaise, and pain [28,29,30,31]. Though most studies are observational, they involve relatively large patient cohorts, and the improvement metrics are significant, with a five-fold higher likelihood of improvement compared to physical therapy alone [28,29,30,31]. Dexamethasone demonstrates clinically relevant reductions in fatigue and shortened symptom duration, with a 33% reduction in fatigue and a shorter median duration of symptoms (133 days vs. 271 days) [32,33]. However, its observational study designs limit the strength of the evidence compared to metformin’s trials [32,33]. Omega-3 fatty acids offer moderate improvements in mental health and musculoskeletal symptoms, such as depression and myalgia [38]. The large patient cohort (33,908 patients) adds weight to the findings, but the retrospective design reduces the quality of evidence [38]. Clinical outcomes are moderate but statistically significant [38]. L-Arginine, combined with vitamin C, shows compelling results, with 94.9% of patients reporting the absence of fatigue and 74.2% reporting no dyspnea [39]. However, the study design (nationwide survey) limits the robustness of the findings [39].

Several non-antiviral treatments for Long COVID show potential but are supported by lower-quality evidence compared to leading therapies like metformin, LDN, and dexamethasone. Therapeutic apheresis yielded significant improvements in 70% of 27 patients, with long-lasting symptom relief [40,41]. Similarly, IVIg improved fatigue and cognitive dysfunction in 70–75% of patients in smaller studies [34,35]. NAC improved cognitive function in 67% of 12 patients but had a 25% discontinuation rate due to side effects [2]. Despite some positive outcomes, these treatments need more robust studies to confirm their efficacy. Figure 2 illustrates the impact of these different treatments on Long COVID patients.

### 4.1. Alternative Therapies and Cognitive Impairment

The cognitive issues observed in individuals recovering from infection, including brain fog, PASC, and dysexecutive syndrome, are closely linked and commonly reported [49,50]. Studies using electroencephalography have found that PASC patients often show decreased delta wave activity in brain regions involved in cognitive processing, which aligns with reduced performance on tasks requiring concentration and memory [49]. Additionally, impairments in GABAergic circuits, especially within the primary motor cortex, have been noted in these patients, connecting these disruptions to both cognitive and physical fatigue [50]. This ongoing neuroinflammatory activity appears to play a central role, not only in perpetuating cognitive deficits similar to those seen in dysexecutive syndrome but also in contributing to widespread dysregulation across brain networks, which is a characteristic of the brain fog frequently reported in these patients [49,50].

Guanfacine, an α2A-adrenoceptor agonist, combined with the antioxidant N-acetylcysteine (NAC), has shown potential in addressing brain fog and related cognitive impairments [2,51]. Guanfacine works by strengthening prefrontal cortex connectivity, which is essential for processes such as working memory and concentration, both commonly affected in brain fog [2,51]. NAC, on the other hand, replenishes intracellular glutathione levels, protecting neurons from oxidative stress, which is often elevated in Long COVID [2,52].

Low-dose naltrexone (LDN), typically used for autoimmune disorders, has demonstrated benefits in reducing brain fog and fatigue in Long COVID. Through its immunomodulatory properties, LDN downregulates pro-inflammatory cytokines, helping to mitigate the chronic inflammation that often contributes to cognitive dysfunction [28,29,30,31,53]. Additionally, LDN’s effect on opioid receptors can positively impact mood and alleviate fatigue, creating a holistic improvement in patients’ quality of life [28,29,30,31,53]. In studies, over 50% of participants reported cognitive improvements with LDN, although a small percentage discontinued due to mild side effects such as fatigue and gastrointestinal symptoms [28,29,30,31].

Coenzyme Q10 (CoQ10) offers another promising approach by supporting mitochondrial function, often impaired in Long COVID [3,25]. As a critical element in mitochondrial energy production, CoQ10 helps alleviate symptoms of fatigue and muscle weakness, indirectly aiding cognitive recovery by enhancing physical endurance and reducing oxidative stress [3,25,54]. While specific improvements in neurological symptoms were less pronounced, CoQ10 did show substantial fatigue reduction, which can positively impact cognitive function over time [3,25].

Intravenous immunoglobulin (IVIg) therapy has also demonstrated considerable effectiveness in treating brain fog in Long COVID patients [34,35]. IVIg works by neutralizing autoantibodies and modulating immune responses, effectively reducing the neuro-inflammation that contributes to cognitive dysfunction [34,35,55]. This immune modulation helps alleviate fatigue, brain fog, and muscle weakness, restoring cognitive and physical function to a notable degree in many patients. Similarly, therapeutic apheresis has also shown promise by removing inflammatory cytokines, autoantibodies, and other pathogenic molecules from the blood, helping to reduce immune dysregulation and chronic inflammation that contribute to cognitive dysfunction [40,41,56].

Dexamethasone, a corticosteroid commonly used to reduce inflammation in acute COVID-19 cases, has shown potential in alleviating persistent symptoms of brain fog and fatigue in Long COVID. By dampening systemic inflammation and reducing pro-inflammatory cytokines, dexamethasone indirectly improves cognitive symptoms [32,33,57].

Omega-3 fatty acids, particularly eicosapentaenoic acid (EPA), offer a supportive role in addressing inflammation-related symptoms such as mood and musculoskeletal pain, both of which can impact cognitive clarity [38,58]. EPA reduces inflammation through downregulating pro-inflammatory cytokines and enhancing endothelial function, thus supporting cardiovascular health and promoting better blood flow to the brain [38,58]. These effects contribute to modest cognitive improvements and reductions in depressive symptoms and muscle pain.

Metformin, an antidiabetic medication, has emerged as a preventative treatment for Long COVID symptoms, particularly when administered early [26,27,59]. Its anti-inflammatory properties and immune modulation effects reduce the risk of developing cognitive symptoms by addressing inflammation and oxidative stress at the onset of the disease [26,27]. When initiated early, it prevented the development of prolonged symptoms, indicating its strong protective potential against brain fog and related cognitive issues [26,27]. Lastly, L-arginine may help alleviate brain fog in Long COVID by increasing nitric oxide production, which improves blood circulation and brain oxygenation [39,60]. By enhancing endothelial function and reducing inflammation, L-arginine supports cognitive clarity and helps reduce associated fatigue [39,60]. Table 3 summarizes the impact of different therapies on cognitive impairment in Long COVID patients.

### 4.2. Alternative Therapies and Systemic Health Benefits

Guanfacine has been shown that it inhibits the replication of influenza virus by disrupting processes after viral protein synthesis, demonstrating a pleiotropic antiviral effect [51]. Agonists like guanfacine showed broad-spectrum antiviral activity in vitro by reducing viral titers by up to 95%, although they did not directly reduce virus replication in vivo but improved survival by protecting against lung damage, such as edema [51]. Similarly, NAC demonstrates pleiotropic effects against infections by modulating immune responses, reducing viral replication, and alleviating inflammation [52]. In a study, 600 mg of oral NAC twice daily significantly reduced the occurrence of flu symptoms, with only 25% of infected individuals showing symptoms compared to 79% in the placebo group, highlighting its potential in reducing respiratory infections like COVID-19 [52]. Together, guanfacine and NAC are believed to inhibit synergistically viral replication and thus reduce the occurrence of Long COVID [2].

Clinical studies on naltrexone, originally used at higher doses to treat opioid addiction, have demonstrated that 4.5 mg of LDN daily resulted in a significant reduction in inflammatory markers and symptoms in chronic autoimmune conditions, where over 80% of participants showed improvement, demonstrating its broad therapeutic potential [53]. Its proposed mechanisms include the upregulation of endogenous opioid receptors, which can have a beneficial impact on pain and mood disturbances commonly experienced in Long COVID [28,29,30,31].

CoQ10 as an important antioxidant exhibits pleiotropic functions by also reducing inflammation, which is a critical factor in the progression of infectious diseases [54]. In clinical trials, supplementation of 200 mg/day of CoQ10 significantly reduced TNF-α inflammatory marker, and improved survival in sepsis patients, demonstrating its potential as an adjunct therapy in managing infections [54]. By improving mitochondrial respiration and reducing oxidative damage, CoQ10 may help alleviate these symptoms, although more research is needed to fully elucidate its role in Long COVID treatment [3,25].

IVIg are immunomodulatory agents composed of pooled antibodies from healthy donors. Five day administration of high-dose IVIg has led to significant clinical improvement in COVID-19 patients, including normalization of oxygen saturation and reduction in inflammatory markers, with all patients recovering from severe symptoms [55]. IVIg is also thought to have broader immunoregulatory effects, such as the inhibition of pro-inflammatory chemokines as well as the adaptation of immune cellular activity, making it a promising treatment for autoimmune-related components of Long COVID [34,35].

Therapeutic apheresis as a treatment approach is designed to eliminate harmful substances from the plasma, such as autoantibodies, inflammatory cytokines, lipids, and oxidative molecules [40,41]. In clinical studies, therapeutic plasma exchange reduced 28-day mortality by 20.5% in severe sepsis patients, while other extracorporeal techniques, such as high-volume hemofiltration and hemoadsorption, showed significant reductions in inflammatory markers, though without consistent survival benefits [56]. Despite being invasive and costly, therapeutic apheresis may offer a novel approach to managing severe cases of Long COVID.

Although primarily used for managing severe acute COVID-19, there is interest in the potential for dexamethasone to reduce the long-term inflammation associated with Long COVID [32,33]. In clinical trials, dexamethasone administration lowered mortality by 12.3% in patients on invasive mechanical ventilation and by 4.2% in those on oxygen support, highlighting its significant benefit in severe respiratory cases [57]. By dampening the inflammatory cascade that may contribute to chronic symptoms, dexamethasone could help alleviate persistent respiratory issues and other inflammatory sequelae [32,33].

Omega-3 fatty acids, due to their anti-inflammatory mechanism of action, have been proposed as a treatment for Long COVID to modulate chronic inflammation and improve endothelial function [38]. Additionally, omega-3 fatty acids may enhance cardiovascular health by improving endothelial function and reducing the risk of thrombosis, both of which are relevant to the vascular complications seen in Long COVID [38]. In a study of 100 hospitalized COVID-19 patients, those with higher omega-3 index levels (≥5.7%) had a 75% lower risk of death compared to those with lower levels [58].

Metformin improves insulin sensitivity, reduces oxidative stress, and modulates immune function [26,27]. Additionally, metformin’s role in reducing the risk of cardiovascular events could be beneficial for patients experiencing vascular complications as part of their Long COVID presentation [26,27]. Metformin exhibits pleiotropic effects due to anti-inflammatory activity and inhibition of viral replication [59]. In a study of diabetic patients with influenza A virus, metformin administration decreased influenza-related mortality by 62% [59].

L-arginine, classified as a conditionally indispensable amino acid, plays a pivotal role as the biochemical precursor to nitric oxide [39]. L-arginine demonstrates pleiotropic effects in combating infections, particularly in COVID-19 patients, by reducing inflammation and improving endothelial function [60]. In a randomized clinical trial, L-arginine supplementation significantly decreased the need for respiratory support by 82% after 10 days and reduced hospital stays from 36 to 22 days, while also lowering pro-inflammatory cytokines like IL-6 and increasing anti-inflammatory IL-10 [60]. L-arginine supplementation in Long COVID patients may potentially improve endothelial function and enhance blood flow, particularly in patients with symptoms related to cardiovascular or respiratory dysfunction [39].

Statins, widely known for their cholesterol-lowering properties, have garnered attention for their potential to mitigate cardiovascular complications in Long COVID patients [37]. Beyond their lipid-regulating effects, statins possess anti-inflammatory and immunomodulatory properties that may reduce the systemic inflammation associated with COVID-19. In a large retrospective study involving 13,981 participants, it was demonstrated that in-hospital statin use reduced all-cause mortality by 42%, highlighting their potential to improve outcomes in severe infections [61,62]. Statins appear to decrease the rate of major adverse cardiovascular events (MACE) in Long COVID patients [37]. Interestingly, maraviroc, a CCR5 antagonist, and pravastatin, a statin-targeting fractalkine, have also been investigated as a combination therapy for Long COVID [36]. Participants reported a significant reduction in the fatigue severity score as well as reductions in inflammatory biomarkers, such as IL-2 [36]. By stabilizing atherosclerotic plaques and modulating the immune response, statins help prevent cardiovascular complications that are exacerbated by COVID-19′s pro-inflammatory state [37]. Furthermore, their ability to reduce oxidative stress and endothelial dysfunction may improve overall vascular health, contributing to a reduction in fatigue, muscle weakness, and cognitive impairment in Long COVID patients [37].

### 4.3. Emerging Treatments

Emerging treatments for Long COVID focus on targeting chronic inflammation, immune dysregulation, and hormonal imbalances associated with the condition. Anakinra, an IL-1 receptor antagonist, helps reduce systemic inflammation, potentially alleviating symptoms like fatigue [63,64,65]. It has previously been reported that 25% of COVID-19 patients of the anakinra group resulted in severe clinical outcomes or mortality versus the 73% in the control group, representing a significant reduction in risk [66]. Fluvoxamine, an SSRI with anti-inflammatory properties, could address both lingering inflammation and neuropsychiatric symptoms such as brain fog [67,68]. In the TOGETHER trial, fluvoxamine administration reduced the risk of hospitalization or extended emergency care by 32%, resulting in a relative risk of 0.68 [69]. Infliximab, a TNF inhibitor, is being explored for its potential to modulate the immune response, while Testofen, known for its testosterone-boosting effects, may alleviate fatigue and muscle weakness linked to hormonal imbalances [70,71]. An infliximab/tocilizumab combination reduced mortality rates to 7%, compared to 14.2% in the tocilizumab-only group and 42.5% in the standard treatment group, while also significantly lowering the need for mechanical ventilation and ICU admissions of COVID-19 patients [72]. Low testosterone has also resulted in an increased mortality risk and ICU admission rate, and hypogonadism was prevalent in 89.8% of male patients, highlighting testosterone’s potential role in modulating immune response and inflammation during infections [73]. Pentoxifylline, with its anti-inflammatory and vasodilatory properties, shows promise in addressing circulatory issues and systemic inflammation [74]. In a study involving severe COVID-19 patients, pentoxifylline reduced lactate dehydrogenase by around 30% while it increased lymphocyte levels by over 60%, indicating its potential to alleviate hyperinflammation and improve immune response [75]. Finally, ImmuneRecov™, a nutritional blend, has demonstrated potential in modulating immune responses and managing cytokine-mediated damage in Long COVID patients [76]. The blend significantly reduced pro-inflammatory cytokines, such as IL-6, while increasing anti-inflammatory IL-10, suggesting that it helps modulate cytokine levels and counteract the hyperactivation caused by SARS-CoV-2, potentially reducing COVID-19 progression and sequelae [76].

Vitamin D (VD), through its interaction with the vitamin D receptor (VDR), has been shown to modulate both the innate and adaptive immune systems, which is critical in combating respiratory viruses like SARS-CoV-2 [77]. Vitamin D plays a role in regulating autophagy, reducing inflammatory cytokine production, and promoting the clearance of viral particles through immune system enhancement [78,79,80]. For instance, a retrospective study involving 2342 hospitalized COVID-19 patients found that serum 25-hydroxyvitamin D deficiency was strongly correlated with a poorer prognosis, including increased mortality and longer ICU stays [77,81]. In terms of Long COVID, some observational studies reported that patients with persistent post-COVID symptoms tended to have lower VD levels than those who recovered without complications [82,83]. For instance, a case–control study with 50 Long COVID patients and 50 control subjects found significantly lower VD levels in the Long COVID group, suggesting a potential role for supplementation in preventing or alleviating these prolonged symptoms [82,83]. Overall, Vitamin D supplementation shows promise for improving outcomes in COVID-19 and Long COVID patients, particularly in those with deficient baseline levels [84,85].

Melatonin is a potent antioxidant and anti-inflammatory agent, with demonstrated immunomodulatory properties [86,87]. Furthermore, its role as a chronobiotic is crucial in restoring circadian rhythms, which can be disrupted in Long COVID, particularly for patients experiencing sleep disturbances or cognitive dysfunction [88,89,90]. Quantitatively, melatonin has been shown to improve cognitive function in other neurodegenerative diseases [86,91,92,93,94,95]. For instance, a retrospective analysis of patients with mild cognitive impairment treated with melatonin showed significant improvements in cognitive performance, with reductions in depression and better sleep/wake rhythms [91,92,93,94,95]. While direct clinical trials on Long COVID are limited, melatonin has shown measurable benefits in other conditions that share symptoms with Long COVID [94,95]

### 4.4. Limitations

This review is subject to several limitations that should be considered when interpreting the findings. First, there is significant heterogeneity among the included studies in terms of design, patient populations, and outcomes, which complicates direct comparisons and limits the generalizability of the results. Second, many of the interventions reviewed are supported by observational studies or small-scale trials, which reduces the strength of the evidence and underscores the need for larger, well-designed RCTs to validate these treatments. Third, the focus of most studies was on short- to mid-term outcomes, with limited data on long-term efficacy, a crucial factor for assessing the sustainability of therapeutic benefits in Long COVID management. Finally, as Long COVID is an emerging condition with evolving clinical understanding, the conclusions drawn from the current body of research may be influenced by new discoveries or viral variants, which could alter the treatment landscape and the applicability of these findings over time.

## 5. Conclusions

The management of Long COVID remains a complex and evolving challenge. This study highlights several promising therapeutic approaches, with metformin emerging as a leading candidate for both the prevention and management of this chronic condition due to its demonstrated efficacy in large, high-quality randomized trials. LDN also shows significant potential, particularly in alleviating symptoms such as fatigue and post-exertional malaise, although most evidence comes from observational studies. Other treatments, including omega-3 fatty acids and emerging therapies such as IVIg and therapeutic apheresis, offer moderate improvements but require further validation through larger, well-designed clinical trials. While antiviral treatments can reduce the progression of Long COVID symptoms if administered early, non-antiviral alternatives may provide broader symptom management by addressing immune dysregulation, inflammation, and other complex pathophysiological aspects associated with Long COVID. The non-antiviral therapies, particularly those addressing immune and inflammatory responses, emerge as valuable for comprehensive management of Long COVID, potentially complementing or enhancing the effects of antiviral treatments, thereby offering a multi-faceted approach that goes beyond early viral suppression.

Given the evolving nature of the pandemic and appearance of new viral strains, future research should also focus on the long-term health impacts of Long COVID and the effectiveness of these therapies across different viral strains and patient demographics. This is particularly important for high-risk individuals, such as the immunocompromised, and patients with pre-existing co-morbidities, to ensure that treatment strategies can be adapted to address their specific needs.

## Figures and Tables

**Figure 1 viruses-16-01795-f001:**
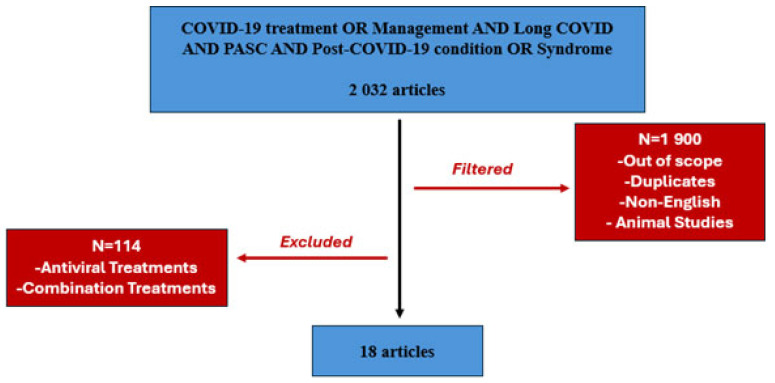
Methodological process for selecting Long COVID articles.

**Figure 2 viruses-16-01795-f002:**
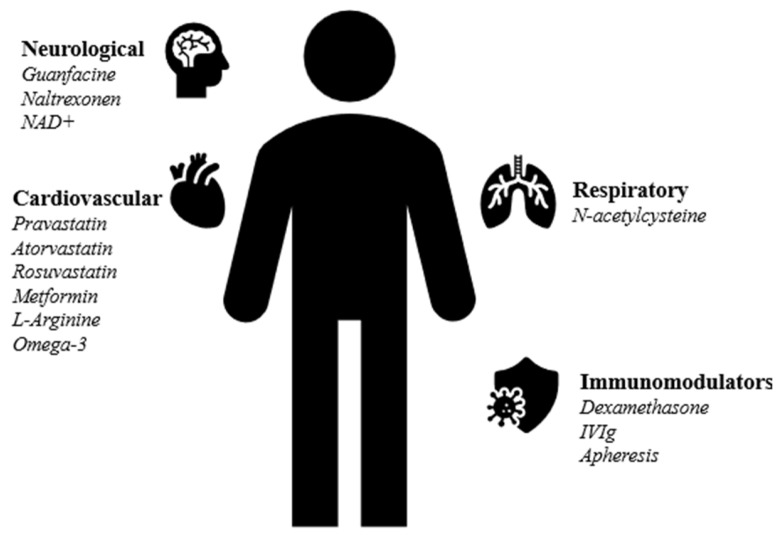
Available treatments for Long COVID patients and main outcomes.

**Table 1 viruses-16-01795-t001:** Representative studies on the impact of different therapeutics on Long COVID.

Author and Country	Patient Population	Study Design	Intervention and Dosage	Outcomes
**Therapeutic Category: Antihypertensive/ADHD Treatment**				
Fesharaki-Zadeh A. et al., 2023 (USA) [2]	12 patients, aged 21–73 years	Open-label clinical experience	Guanfacine: Initiated at 1 mg orally for 1 month, increased to 2 mg after if well tolerated N-acetylcysteine: 600 mg orally daily Arms: Combined treatment No control group	A total of 8/12 reported cognitive improvements such as enhanced working memory, concentration, and executive functions
**Therapeutic Category: Antioxidant**				
Hansen K.S. et al., 2023 (Denmark) [3]	121 patients, aged 22–70 years	Phase 2 RCT	Coenzyme Q10: Daily dose of 500 mg, administered as five 100 mg oral capsules to maximize absorption Dosing period for 6 weeks, followed by a 4-week washout before switching treatments in a crossover design Arms: Coenzyme Q10 vs. placebo	The largest non-significant reduction in symptoms was in neurological and musculoskeletal symptoms, but it was not statistically significant
Barletta M.A. et al., 2023 (Italy) [25]	174 patients, aged 18–81 years	Prospective observational study	100 mg of alpha lipoic acid/Coenzyme Q10 each twice daily for 2 months Arms: 116 patients on combination vs. 58 patients control	A total of 53.5% of treated patients achieved complete response (a ≥ 50% reduction in Fatigue Severity Scale) vs. 3.5% in the control
**Therapeutic Category: Antidiabetic**				
Bramante C.T. et al., 2024 (USA) [26]	999 patients, aged 30–85 years	Phase 3 RCT	Metformin: 500 mg on the first day, 500 mg twice until day five, and 500 mg or 1000 mg morning/evening, respectively, until day 14 Fluvoxamine: 50 mg the first day, and 50 mg twice daily until day 14 Ivermectin: A median dose of 430 µg/kg/day for 3 days Arms: Each drug was compared to a placebo group	Long COVID reduction by 42% due to metformin Long COVID 65% reduction if metformin treatment started within four days
Bramante C.T. et al., 2023 (USA) [27]	1431 patients, aged 30–85 years	Phase 3 RCT	Metformin: 500 mg on the first day, 500 mg twice until day five, and 500 mg or 1000 mg morning/evening, respectively, until day 14 Ivermectin: Up to 470 μg/kg daily for 3 days. Fluvoxamine: Once 50 mg the first day, and 50 mg twice daily until day 14 Arms: Each drug was compared to a placebo group	Long COVID metformin reduced the rate 41% Until day 300 of 6.3% vs. 10.4% for controls developed Long COVID Administered within 3 days of symptom onset, it provided an even greater reduction in long COVID risk, with an HR of 0.37 (95% CI 0.15–0.95)
**Therapeutic Category: Immunomodulator**				
Bonilla H. et al., 2023 (USA) [28]	59 patients, aged 34–59 years	Retrospective observational study	Low-dose naltrexone: Average of 2 mg daily, median duration of treatment was 143 days Arms: Combined treatment No control group	Fatigue significant reduction (*p* = 0.013). Post-exertional malaise improved (*p* = 0.010). Unrefreshing sleep and abnormal sleep patterns were significantly improved (*p* = 0.010 and *p* = 0.016, respectively)
Isman A. et al., 2024 (USA) [29]	36 patients, aged 28–69 years	Observational, open-label pilot study	Low-dose naltrexone: 4.5 mg/day, the dose was gradually increased over the first 9 days NAD+: Using iontophoresis patches, delivering 400 mg of NAD+ weekly Interventions lasted 12 weeks Arms: Combined treatment No control group	SF-36 score increased from 36.5 to 52.1 at 12 weeks, showing a significant improvement (*p* < 0.0001) Both physical and mental fatigue scores showed substantial improvement
Tamariz L. et al., 2023 (USA) [30]	108 patients, aged	Retrospective cohort study	LDN, amitriptyline, duloxetine Arms: LDN vs. physical therapy	Relative hazard of improvement for LDN group was 5.04 (95% CI: 1.22–20.77) vs. control Fatigue and pain were improved for LDN group
O’Kelly B. et al., 2022 (Ireland) [31]	52 patients, aged 33–49 years	Interventional study	1 mg of LDN daily in the first month and 2 mg daily in the second month. Arms: No control group	Statistical improvement in multiple metrics including fatigue levels, measurements of pain, cognitive performance, and sleep quality
Milne A. et al., 2023 (UK) [32]	198 patients, aged 51–72 years	Observational case–control study	Dexamethasone: 39 patients received 6 mg of oral dexamethasone once daily during their hospital stay, median duration of treatment was 7 days Arms: Dexamethasone vs. control	Most significant reductions (8-month follow-up) were in fatigue (33% for dexamethasone vs. 52% in for placebo, *p* = 0.07) and insomnia (15% in the dexamethasone group vs. 39% in control group, *p* = 0.01)
Badenes Bonet D. et al., 2023 (Spain) [33]	1966 patients, mean age 56.4 years	One-year prospective observational study	Dexamethasone: 6 mg daily for 10 days Arms: Dexamethasone vs. control	The median duration for symptoms for patients treated with dexamethasone was 133 days (compared to 271 days for those not treated with dexamethasone)
Hogeweg M. et al., 2023 (Germany) [34]	30 patients	Retrospective case–control study	Intravenous immunoglobulins: 3–4 monthly courses of IVIg, administered at a dose of 0.5 g/kg (Group A) Inhaled glucocorticoids: budesonide inhalation at a dosage of 0.2 mg twice daily (Group B) Supportive Therapy: Mon-pharmacological treatments managing post-COVID symptoms (Group C) Arms: No combinations and compared to supportive therapy	Symptom Improvement (ISARIC Score): Change in score was significantly higher in group A (−5.44 ± 2.35; *p* < 0.001) compared to Group B (−0.3 ± 0.82) and Group C (−0.89 ± 0.93)
Thompson J.S. et al., 2022 (USA) [35]	9 patients, aged 34–79 years	Observational case study	The standard treatment involved 0.5 g/kg of IVIg bi-weekly for a trial period of 3 months. Arms: IVIg vs. untreated patients	Six patients received IVIg treatment, and all reported significant to remarkable improvements, particularly in cognitive dysfunction, fatigue, and pulmonary issues
**Therapeutic Category: Statin**				
Khosravi A et al., 2023 (Iran) [37]	858 patients, aged 18–85 years	Prospective cohort study	Atorvastatin 20 mg daily or Rosuvastatin 10 mg daily for at least 15 days Arms: Statin vs. control	MACE risk After median follow-up of 13 months statins (HR: 0.831; 95% CI: 0.529–0.981)
**Therapeutic Category: Other**				
Liu T.H. et al., 2023 (China) [38]	33,908 patients, aged 18–85 years	One-year retrospective cohort analysis	Omega-3 polyunsaturated fatty acids supplements Omega-3 intake was within 6 months prior to their COVID-19 diagnosis Arms: Omega-3 vs. no use	Depression: HR = 0.828; (95% CI:0.714–0.960) Myalgia: HR = 0.606; (95% CI: 0.417–0.880) Cough: HR = 0.814; (95% CI: 0.683 to 0.970)
Izzo R. et al., 2022 (Italy) [39]	1390 patients, aged 18–90 years	Nationwide survey-based observational study	L-Arginine: 1.66 g (two vials/day). Vitamin C: 500 mg of liposomal Vitamin C daily. Patients in the alternative treatment group received a multivitamin combination Arms: Combined treatment. No control group	Fatigue: Absent in 94.9% of patients in the L-Arginine + Vitamin C group compared to 0.4% in the control group Dyspnea: Absent in 74.2% of patients in the L-Arginine + Vitamin C group compared to 5.4% in the control group
Jaeger B.R. et al., 2023 (Germany) [40]	17 patients, aged 23–63 years	Pilot study	H.E.L.P. Apheresis: In each session, 400,000 units of unfractionated heparin were used in the process During each treatment, between 2 and 4 L of blood were processed. The treatment lasted between 2 and 4 h per session Arms: No control group	A total of 16 patients experienced significant symptom improvement Follow-up between 6 and 10 months after the last treatment, 15 patients maintained their improvements
Achleitner M. et al., 2023 (Germany) [41]	27 patients Mean age of 49.7 years for men and 44.9 years for women	Prospective observational study	Therapeutic Apheresis: Each session lasted 114 min on average, and 8000 units of heparin were used per treatment. Arms: No control group	A total of 70% of the patients reported a significant improvement in symptoms such as fatigue, post-exertional malaise, and brain fog

IVIg: intravenous immunoglobulins; LDN: low-dose naltrexone; RCT: Randomized Controlled Trial; MACE: Major Adverse Cardiovascular Event; HR: Hazard Ratio; CI: Confidence Intervals.

**Table 2 viruses-16-01795-t002:** Advanced, in-progress pharmacotherapy trials on Long COVID.

Trial Number and Estimated Completion Date	Patient Population	Study Design	Intervention and Dosage	Outcome Measure
**Therapeutic Category: Immunomodulator**				
NCT05926505 (August 2025) [42]	182 patients 18 years and older	Phase 2–3 RCT	Anakinra injected subcutaneously as 100 mg once daily for 4 weeks Placebo injected subcutaneously once daily for 4 weeks	Score of PACS progression reversal Changes in concentration of cytokines produced
NCT05513560 (December 2025) [43]	1000 patients 18 years and older	Phase 2–3 RCT	Ibudilast 10 mg pills, 2 pills twice per day Pentoxifylline 400 mg pill 3 times per day Placebo matching ibudilast OR pentoxifylline dosing schedule	SF-36 physical component score
NCT05220280 (December 2025) [44]	400 patients 18 years and older	Phase 4, Randomized, Open Label	Oral imatinib 400 mg tablet once a day for 14 days Infliximab as a single infusion of 5 mg/kg	Health-related quality of life Mortality
**Therapeutic Category: Anti-depressant/Antidiabetic**				
NCT05874037 (May 2025) [45]	300 patients 25 years and older	Phase 2–3 RCT	An amount of 25 mg/50 mg/100 mg fluvoxamine doses to assess impact on each patient Customized dose of fluvoxamine vs. placebo for 16 weeks	Patient-reported assessments on improvement of Long COVID symptoms Biomarkers of underlying inflammatory pathophysiology
NCT06128967 (May 2025) [46]	1500 patients 18 years and older	Phase 3 RCT	Fluvoxamine Maleate 100 mg per pill Oral 750 mg Metformin extended release Placebo matching dosing schedules	Improvement on Fatigue Severity Score Scale
**Therapeutic Category: Other**				
NCT05795816 (November 2025) [47]	150 patients 18 years and older	Phase 3 RCT	Testofen: Twice daily 300 mg dose Microcrystalline cellulose: Same dosing schedule as testofen	Change in energy and fatigue
NCT06166030 (December 2024) [48]	58 patients 18 years and older	Phase 3 Non-Randomized, Open Label	One month administration of ImmuneRecov	Lung function tests and immune reaction

RCT: Randomized Controlled Trial.

**Table 3 viruses-16-01795-t003:** Treatments for Long COVID impacting cognitive symptoms.

Treatment	% Improvement in Cognitive Symptoms	Mechanism
Guanfacine + NAC	67%	Strengthens prefrontal cortex connectivity, reduces oxidative stress
LDN	54%	Modulates immune response, reduces chronic inflammation
CoQ10	Moderate, non-significant improvement in fatigue	Supports mitochondrial function, reduces oxidative stress
IVIg	70–75%	Neutralizes autoantibodies, reduces neuro-inflammation
Omega-3 Fatty Acids	Modest, non-significant improvement in mood, myalgia	Reduces inflammation, supports endothelial function
Dexamethasone	33% reduction in fatigue	Reduces systemic inflammation
Metformin	Preventative (up to 63% reduction)	Reduces inflammation, modulates immune response
Therapeutic Apheresis	70% reported symptom improvement	Removes inflammatory cytokines and autoantibodies, reduces immune dysregulation
L-Arginine	94.9% absence of fatigue, improvement in oxygen delivery	Enhances nitric oxide production, improves endothelial function, reduces inflammation

NAC: N-acetylcysteine; LDN: low-dose naltrexone; CoQ10: Coenzyme Q10; IVIg: intravenous immunoglobulin.

## Data Availability

No new data were created or analyzed in this study. Data sharing is not applicable to this article.

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
