# Peer review of "Beyond Antivirals: Alternative Therapies for Long COVID"

_viruses, 2024, doi:10.3390/v16111795_

Round 1

Reviewer 1 Report

Comments and Suggestions for Authors

ABSTRACT

After introducing the definition of Long Covid-19, I suggest the authors add a general sentence about the treatment of PASC.

Row 10: Please delete the comma before “, is..”

INTRODUCTION

Rows 30-31: I suggest the authors merge this sentence with the following in row 33 “…and is often associated with a more severe initial illness” to avoid a contradictory sentence.

 I suggest the authors add some sentences about Neuro Long-Covid, which includes cognitive difficulties and fatigue, commonly referred to as Brain Fog. Brain Fog is the most prevalent complaint of Long Covid-19 syndrome.

METHODS

Row 98: I suggest the authors use a period as a thousand separator for numbers throughout the text.

Row 101: While “off-topic” and “out of scope” (Figure 1) may seem similar, they are distinct concepts. Please, specify more clearly the inclusion criteria.  

Figure 1: I suggest the authors use the PRISMA flow diagram.

RESULTS

Table 1: I suggest the authors standardize the content of the “Patient Population” column across studies, considering mandatory inclusions such as patient number and age range.

Table 2: I suggest the authors add the age range in the “Patient Population” column.

DISCUSSION

In my opinion, the “Discussion” paragraph would benefit from some thoughtful considerations by the authors, because as it stands, it is very descriptive but not enough theoretical as a discussion should be. In particular, there is a noticeable absence of an in-depth discussion on brain fog (cognitive impairments and fatigue), the primary complication of Long COVID-19 syndrome, even though the findings suggest these drugs may be effective in treating this condition.

Moreover, I suggest the authors improve the summarization of results to ensure clarity and coherence, organizing the information according to a logical flow and/or a specific aim (e.g., the targets of the treatments or the outcomes).

Rows 158-165: I suggest the authors move this part at the start of the discussion, before “Among the available…” (Row 141).

Rows 168-171: Specifically, a relation between Brain Fog, PASC, and dysexecutive syndrome has been found in the literature.

Versace, V.; Sebastianelli, L.; Ferrazzoli, D.; Romanello, R.; Ortelli, P.; Saltuari, L.; D’Acunto, A.; Porrazzini, F.; Ajello, V.; Oliviero, A.; et al. Intracortical GABAergic dysfunction in patients with fatigue and dysexecutive syndrome after COVID-19. Clin.Neurophysiol. 2021, 132, 1138–1143.

Ortelli, P.; Quercia, A.; Cerasa, A.; Dezi, S.; Ferrazzoli, D.;Sebastianelli, L.; Saltuari, L.; Versace,V.; Quartarone, A. Lowered Delta Activity in Post-COVID-19 Patients with Fatigue and Cognitive Impairment. Biomedicines 2023, 11,2228. https://doi.org/10.3390/biomedicines11082228

 Figure 2: The figure aims to provide a concise overview of the findings. Nevertheless, I recommend revising it for greater clarity. For example, eliminating the central human figure would improve the visualization, as there appears to be an inconsistency between the drugs and the body's organs. Furthermore, does the visual summary pertain to the drug's outcome or target?

Rows 231-232: The sentence seems to be interrupted.

Rows 234-235: I suggest the authors move this sentence before “Dexamethasone, a corticosteroid, which …..”.

The Discussion should be carefully proofread for formatting issues.

Limitations

I suggest the authors move this paragraph to the end, after the Conclusion paragraph.

Conclusions: I suggest the authors add some comparisons with the antiviral treatment of Long COVID-19 syndrome, highlighting the strength of alternative drugs to treat this syndrome.

Author Response

Reviewer 1:

ABSTRACT

After introducing the definition of Long Covid-19, I suggest the authors add a general sentence about the treatment of PASC.

Row 10: Please delete the comma before “, is..”

 Thank you for your comments. Both comments have been addressed in the abstract. 

INTRODUCTION

Rows 30-31: I suggest the authors merge this sentence with the following in row 33 “…and is often associated with a more severe initial illness” to avoid a contradictory sentence.

 I suggest the authors add some sentences about Neuro Long-Covid, which includes cognitive difficulties and fatigue, commonly referred to as Brain Fog. Brain Fog is the most prevalent complaint of Long Covid-19 syndrome.

Thank you for your comments. Both comments have been addressed in the abstract. Comment 1 by merging the sentences as requested and Comment 2 by adding some more content on Neuro Long-Covid.

METHODS

Row 98: I suggest the authors use a period as a thousand separator for numbers throughout the text.

Row 101: While “off-topic” and “out of scope” (Figure 1) may seem similar, they are distinct concepts. Please, specify more clearly the inclusion criteria.  

Figure 1: I suggest the authors use the PRISMA flow diagram.

Thank you for your comments. Comment 1 has been addressed throughout the text regarding the separator. Comment 2 the terminology has been aligned between the text and Figure 1 to maintain consistency. Comment 3, Figure 1 has been updated accordingly.

RESULTS

Table 1: I suggest the authors standardize the content of the “Patient Population” column across studies, considering mandatory inclusions such as patient number and age range.

Table 2: I suggest the authors add the age range in the “Patient Population” column.

Thank you for your comments. Table 1 has updated the ‘‘Patient Population’’ column and standardized as requested. However, not all studies currently have the requested information and as a result when information was missing, a mean value might have been provided regarding ages, where available. Table 2 there is no further available information to add. Studies focused on adult populations.

DISCUSSION

In my opinion, the “Discussion” paragraph would benefit from some thoughtful considerations by the authors, because as it stands, it is very descriptive but not enough theoretical as a discussion should be. In particular, there is a noticeable absence of an in-depth discussion on brain fog (cognitive impairments and fatigue), the primary complication of Long COVID-19 syndrome, even though the findings suggest these drugs may be effective in treating this condition.

Moreover, I suggest the authors improve the summarization of results to ensure clarity and coherence, organizing the information according to a logical flow and/or a specific aim (e.g., the targets of the treatments or the outcomes).

Thank you for your comments. The Discussion has now been segregated into 3 sections and organized to ensure coherence and logical flow. Furthermore, a section focusing on the in-depth discussion on the Neuro Long Covid aspect has been included. Additionally, to further address this comment an extra table has been included summarizing findings on this topic.

Rows 158-165: I suggest the authors move this part at the start of the discussion, before “Among the available…” (Row 141).

Thank you for your comment. The structure of this section has remained the same. This is because treatments with best supporting evidence are logically presented first with less promising findings afterwards.

Rows 168-171: Specifically, a relation between Brain Fog, PASC, and dysexecutive syndrome has been found in the literature.

Versace, V.; Sebastianelli, L.; Ferrazzoli, D.; Romanello, R.; Ortelli, P.; Saltuari, L.; D’Acunto, A.; Porrazzini, F.; Ajello, V.; Oliviero, A.; et al. Intracortical GABAergic dysfunction in patients with fatigue and dysexecutive syndrome after COVID-19. Clin.Neurophysiol. 2021, 132, 1138–1143.

Ortelli, P.; Quercia, A.; Cerasa, A.; Dezi, S.; Ferrazzoli, D.;Sebastianelli, L.; Saltuari, L.; Versace,V.; Quartarone, A. Lowered Delta Activity in Post-COVID-19 Patients with Fatigue and Cognitive Impairment. Biomedicines 2023, 11,2228. https://doi.org/10.3390/biomedicines11082228

Thank you for your comment. Further information has been added in the Discussion Section- Alternative Therapies and Cognitive Impairment- and these references now included in the manuscript.

Figure 2: The figure aims to provide a concise overview of the findings. Nevertheless, I recommend revising it for greater clarity. For example, eliminating the central human figure would improve the visualization, as there appears to be an inconsistency between the drugs and the body's organs. Furthermore, does the visual summary pertain to the drug's outcome or target?

Thank you for your comment. The Figure has been updated to identify the drug’s main effects.

Rows 231-232: The sentence seems to be interrupted.

Rows 234-235: I suggest the authors move this sentence before “Dexamethasone, a corticosteroid, which …..”.

The Discussion should be carefully proofread for formatting issues.

Thank you for the comments. As part of the wider re-organization addressed in a previous comment on the Summarization/Flow of the Discussion section these comments are addressed as the interrupted sentence is deleted.

Limitations

I suggest the authors move this paragraph to the end, after the Conclusion paragraph.

Thank you for the comment. Structure has remained unaltered as Conclusions paragraph has been enriched as suggested on the next comment for concluding the manuscript.

Conclusions: I suggest the authors add some comparisons with the antiviral treatment of Long COVID-19 syndrome, highlighting the strength of alternative drugs to treat this syndrome.

 Thank you for the comment. This comparison aspect has been added in this conclusion section.

Reviewer 2 Report

Comments and Suggestions for Authors

The authors' review provides an overview of the current evidence on non-antiviral treatments for Long COVID, highlighting potential therapeutic options and identifying areas where further research is urgently needed. The manuscript is well written. Overall, this work significantly contributes to the understanding of alternative therapies for long-term COVID and its potential clinical applications. Before publication, authors should check for plagiarism and minor grammar corrections.

Comments on the Quality of English Language

NA

Author Response

Reviewer 2:

The authors' review provides an overview of the current evidence on non-antiviral treatments for Long COVID, highlighting potential therapeutic options and identifying areas where further research is urgently needed. The manuscript is well written. Overall, this work significantly contributes to the understanding of alternative therapies for long-term COVID and its potential clinical applications. Before publication, authors should check for plagiarism and minor grammar corrections.